# Is pedagogical training an essential requirement for inclusive education? The case of faculty members in the area of Social and Legal Sciences in Spain

Rosario Lopez-Gavira[1]☯*, Inmaculada Orozco[2]☯, Ana Doménech[3]☯

1 Department of Accounting, Faculty of Economics and Business, Universidad de Sevilla, Sevilla, Spain,
2 Department of Teaching and Educational Organization, Faculty of Education, Universidad de Sevilla, Sevilla, Spain, 3 Department of Didactics, Faculty of Education, Universidad de Cadiz, Cadiz, Spain

☯ These authors contributed equally to this work.
* rlopezgavira@us.es

**Data Availability Statement:** All relevant data are within the manuscript and its Supporting information files.

## Abstract

### Introduction

This study investigates the profiles of 25 faculty members of the area of Social and Legal Sciences, from seven Spanish universities, who were selected by students with disabilities. Specifically, the aim of this paper is to understand why, despite their lack of pedagogical and disability training, they are considered to be inclusive faculty members.

### Objective

For this purpose, we analysed the professional characteristics of these academics, their conceptions about disability, what actions they carry out to provide an inclusive response and how they consider that the current situation of university students with disabilities could be improved.

### Methods

Semi-structured individual interviews were used to collect the data. Subsequently, a progressive analysis of the data was performed, using an inductive system of categories and codes.

### Results and conclusions

The results show that these faculty are responsible, involved and committed to their teaching performance. Moreover, they regard reasonable adjustments as a fundamental requirement to handle the different needs of students with disabilities. This paper–which can also be useful for faculty and researchers in other areas of knowledge–comes to the conclusion that training is relevant for becoming an inclusive faculty member. Nevertheless, good will and eagerness to do one's job properly are even more important aspects.

**Funding:** This work was supported by the Ministry of Economy and Competitiveness of Spain and ERDF funds [grant numbers EDU2016-76587-R]. https://portal.mineco.gob.es/es-es/Paginas/default.aspx NO.

**Competing interests:** The authors have declared that no competing interests exist.

## Introduction

There are different activities in Higher Education (HE) to ensure that students with disabilities are treated under conditions of equity. The right to inclusive university education is gathered in different legislative activities. At international level, specifically in Article 24 of the Convention on the Rights of Persons with Disabilities [1], it is indicated that all countries must respect, protect and guarantee an inclusive and quality education for all people, without exceptions. The Spanish legislative framework also recognises the universal right to inclusive education. In particular, it is specified that students must not be discriminated due to disability regarding access, admission, permanence and attainment of academic titles [2].

In Spain, the number of students with disabilities in all knowledge areas and, particularly, in Social and Legal Sciences, has increased exponentially [3]. The Universia Foundation report [3] states that 54.5% of the 21,435 university students with disabilities who were registered in the last academic year of which data had been gathered (2017–2018) were enrolled in Social and Legal Sciences.

However, people with disabilities are still facing barriers to access university and be recognised as holders of rights [4]. Moreover, the proportion of students with disabilities who managed to complete their university studies was lower compared to the rest [5, 6]. This indicates that inclusion is still a pending task [7, 8] and largely dependent on the good will of the institutions and university staff [9].

Different studies assert that the medical model of disability is still very present in the scope of HE [10, 11]. This implies that disability is perceived as an individual problem. It is essential to avoid this perception and apply changes in order to adopt a more humane perspective as established by the social model of disability [8, 12]. This model considers that the causes of disability are social and that the limitations of the person result from the lack of support or the barriers imposed by society to their inclusion [13]. Therefore, rather than a problem, disability is an opportunity for both faculty (to become better professionals and learn new methodological strategies, skills and patience) and students without disabilities (to be inspired by effort and motivation of their fellow students and to become more sensitive toward disability in general) [14, 15].

In this study, we advocate the social model of disability and inclusive pedagogy as a socially fair alternative [4, 16]. Thus, inclusive pedagogy is an approach of teaching and learning that supports educators in responding to the individual differences among students, while preventing the potential marginalisation that could emerge when some students are treated in a different manner [17]. In this context, it is convenient to plan the teaching practice taking into account all students from the beginning. This is best achieved by the Universal Design for Learning (UDL) approach, which is a didactic method that promotes different means of representation (the "what" of learning), action and expression (the "how" of learning) and involvement (the "why" of learning) to ensure that all students can access the curriculum [18]. Therefore, the benefits of this approach are there for everyone and not only for students with disabilities [19].

To make these practices possible, it is fundamental to bear in mind that faculty is a key element for the inclusion of students [20, 21]. Thus, it is important to know the characteristics of inclusive faculty members.

Several studies indicate that inclusive faculty members are characterised by showing a positive attitude toward disability [16, 22], maintaining close relationships with their students [8, 23] and using innovative teaching methodologies that engage students [24]. Regarding disability, inclusive faculty members show good predisposition to make reasonable adjustments [9, 25]. To this respect, they value the existence of support services of the universities and the help provided by them. However, in their opinion, such services still require improvements [26, 27].

Despite their positive attitude and the good intentions, faculty members state that, in some cases, they find difficulties to develop inclusive pedagogy due to insufficient training in disability [12, 28]. This aspect is especially sensitive in the area of Social and Legal Sciences, since faculty are responsible and carry out their work with vocation, but they lack a generic base of pedagogical training [29–31].

Literature shows that sufficient teaching resources [12, 25] and adequate faculty training [16] are key to achieve true inclusion of all students. In this sense, several studies in the area of Social and Legal Sciences highlight the issue of student discrimination, the lack of faculty training and the insufficiency of curricular adjustments [30–33]. However, the intrinsic characteristics of Social and Legal Sciences faculty have not been explored, from the perspective of these academics, as key factors of their attention and care for students with disabilities and as important elements to compensate for the lack of training and available resources in HE.

This is an important topic, since the main barriers that students with disabilities encounter are not related to the absence of resources, but to beliefs and culture, that is to say, to how the university understands and welcomes disability [34]. Although addressing disability is a complex undertaking, it can provide the chance to revise and correct teaching decisions and actions with the aim of improving inclusion [11].

Thus, all of the above leads us to explore and understand the particular characteristics of Social and Legal Sciences faculty members who, despite their lack of pedagogical training, are considered as inclusive faculty member by their students with disabilities for having contributed to the inclusion of the latter in the university. Likewise, this study also analysed how these faculty members understood disability, what actions they carried out to provide an inclusive response to their students and what aspects could be improved to make both the teaching practice and the university more inclusive.

## Materials and methods

The results of this qualitative study are framed within a broader research project funded by the Spanish Ministry of Economy and Competitiveness entitled "*Inclusive pedagogy in the university: faculty members' narratives* (EDU2016-76587-R, IP: Anabel Moriña, 2016–2021)". The aim of this project is to determine what, how and why faculty members of different fields of knowledge (Arts and Humanities, STEM, Health Sciences, Social and Legal Sciences) carry out an inclusive pedagogy.

Specifically, this article is focused on analysing why Social and Legal Sciences faculty members, despite their lack of pedagogical training regarding disability, are considered by their students as inclusive faculty. Four research questions guided this investigation:

1. Which are the characteristics of these faculty members that make them inclusive?

2. What is their conception about disability and what motivates them to be inclusive?

3. What actions do they use to provide a response to the students?

4. How can the inclusion of students with disabilities be improved in the field of Social and Legal Sciences?

### Participants

A total of 119 faculty members of different fields of knowledge from 10 Spanish universities participated in this study, of whom 35 taught in the area of Social and Legal Sciences. However,

**Table 1. Participant's profile.**

| Criteria | | n | % |
|---|---|---|---|
| Gender | Male | 14 | 56% |
| | Female | 11 | 44% |
| Age | 30–40 | 3 | 12% |
| | 41–50 | 10 | 40% |
| | 51–60 | 11 | 44% |
| | +61 | 1 | 4% |
| Years of teaching experience | 0–10 | 7 | 28% |
| | 11–20 | 8 | 32% |
| | 21–30 | 7 | 28% |
| | 31–40 | 3 | 12% |
| Fields | Economics and Business | 15 | 60% |
| | Law | 5 | 20% |
| | Journalism | 3 | 12% |
| | Social Work | 2 | 8% |
| Type of disability (of the participants' total experiences with students with disabilities = 43) | Physical | 11 | 25,5% |
| | Sensorial (visual and hearing) | 9 | 20,9% |
| | Psychological disorders | 12 | 27,9% |
| | Learning difficulties | 8 | 18,6% |
| | Organic | 3 | 6,9% |

due time constraints and personal circumstances of the participants, the final sample was constituted by 25 faculty members from 7 Spanish universities (Table 1).

The disability services of the participating universities sent the information of the project to the students with disabilities of all fields of knowledge and asked for their voluntary collaboration. In addition, the snowball technique was used [35]. This way, participation was requested from these students, who were known to the research team for having participated in previous projects. Similarly, other acquainted faculty members who had had experiences with students with disabilities were also asked to participate.

The selection of participants was performed through students with disabilities, who nominated faculty members that, from their experience, had contributed to their inclusion in the university. To support them with this task, the research team sent them, via e-mail, a list of the characteristics that these faculty members should have: 1) facilitate the learning process; 2) promote active learning; 3) use different methodologies; 4) care about the learning of all students; 5) be flexible and help students who need assistance; 6) motivate their students; 7) achieve participation and group learning; and 8) make every student feel important.

## Data collection instruments

Semi-structured individual interviews were used as data collection instruments. With the aim of creating a familiar environment with the participants and offering them flexibility, the research team decided to carry two individual interviews. The first interview was focused on the beliefs and knowledge of the participants, whereas the second interview was focused on their teaching designs and actions. This interview script was tested by faculty members who did not participate in the study. Specifically, here are some of the questions addressed in the results of this article:

- Could you tell me some characteristics that define you as a faculty member? How do you think your attitude or characteristics influence the students' learning?

- When talking about disability, what ideas come to your mind? What do you know about disability? What do you know about inclusive education? What training do you think you should have in order to teach students adequately, especially students with disabilities? What do you think drove you to become interested in and work for the inclusion of students with disabilities?

- When you learn that there is a student with a disability in your subject, what actions do you take? What steps do you follow?

Each full interview lasted approximately 90 minutes, and most of them were conducted face-to-face, with 4 and 3 interviews being performed via telephone and Skype, respectively, due to either personal circumstances, preference or lack of time of the participants.

### Data analysis

Following the proposal of Miles and Huberman [36], all information was transcribed by the research team and then qualitatively analysed using an inductive system of categories and codes. To facilitate the information processing, the data analysis software MaxQDA 12 was employed.

Data analysis consisted of two phases. The first one was a coding phase, characterised by a broad and generic system of categories. The second phase generated new subcodes related to the fundamental topics and ideas. Each subcode was thoroughly analysed for possible decomposition or merging with other codes (Table 2).

All information was categorised, discussed and analysed by the research team, organising the doubtful information.

**Table 2. Categories and codes system.**

| Categories | Codes |
|---|---|
| Characteristics of the faculty members | Responsible, vocational, communicative, empathetic, open, accessible, demanding. |
| Conception about disability, inclusive education and motivations | • Disability: Doubts to define it, awareness about the existence of different types of disability, medical model *versus* social model.<br>• Inclusive education: Lack of pedagogical training, integrative perspective and inclusive perspective.<br>• Motivations: Ethical obligation, professional commitment and fundamental rights. |
| Inclusive actions | Applying common sense, knowing the specific needs, requesting information (students, disability services, external professionals, colleagues), making reasonable adjustments and promoting tutorial action. |
| Aspects to improve | • Faculty: Better coordination with the disability services, promoting communication and trust connections with the students and adopting a proactive attitude.<br>• University: Promoting a regulated faculty training, improving coordination and creating support between the faculty and the students. |

### Ethical matters

This study was approved by the Spanish Ministry of Economy and Competitiveness.

The participants were informed from the beginning of the study through a written informed consent, which they signed and handed to the research team. This document explained the purpose of the study and their rights as participants. Moreover, it guaranteed their anonymity with numbers (Faculty 1-Faculty 25), their approval to be recorded, their voluntary participation and the possibility of leaving the study whenever they wished.

Each transcription was e-mailed to its corresponding participant, who was given the opportunity to make changes in the text.

## Results

The results are organised in four sections, responding to the research questions: Which are the characteristics of the selected faculty? What is their conception about disability? What actions do they take in their classrooms? What aspects could be improved to achieve the true inclusion of students with disabilities?

### The characteristics of faculty members as the background of their inclusive profile

First, it is necessary to assert that all the participants defined themselves as responsible faculty members committed to their subjects. They tried to comply with the entire programme established and were consistent in the different tasks they had to perform (updating contents, tutoring, lecturing, etc.).

Some of them pointed out that the teaching profession should be vocational, stating that they had chosen it consciously, knowing what it entailed and required, without considering other matters, such as the salary. However, they also considered that it was fundamental to communicate properly. They defined themselves as communicative people who knew how to transmit knowledge to the students, connect with them and maintain the attention of the latter with their way of teaching.

*I think that I communicate very well with my students. The knowledge we have is not as important as how we transmit it and ensure that it reaches our students*

(Faculty 19).

Most of them expressed that they were sensitive toward their students in their day-to-day routine, understood their situation, cared for their lives, knew their names and connected with their particularities. They perceived themselves as accessible people who helped each student feel confident to come to them whenever they needed help.

*I believe I am an open and accessible person. I always suggest them to come to the tutorials if there is something they do not understand. When I explain something, I try to get them to confirm that they understood*

(Faculty 14).

Despite this, these characteristics were not in conflict with the requirement level of the subject. The majority of participants considered that, in addition to being open, they were also demanding in terms of academic performance. They believed that students had to face a very

specific work environment and that their lectures had to contribute to preparing them for the professional reality that was awaiting them.

All these characteristics of the participating faculty members influenced the learning of their students, since, in their opinion, they encouraged the students to learn, get involved in the subject and be constantly motivated. From their perspective, this was associated with the organisation and design of the subject itself, which was known in advance by the students.

*I believe that I provide that*: *anticipation and confidence. All this catches them and allows them to work in the subject with constant motivation* (Faculty 25). In short, their teaching responsibility, demanding nature and commitment to the learning of their students with disabilities made the latter also feel comfortable with the subject, confident and sure about what they were learning.

## The concept of disability as a motivation for being an inclusive faculty member

Most of the participants highlighted that they did not have previous training or knowledge about disability. In fact, some of them did not even want to define it, out of fear of not using the right words.

*I would find it difficult to define what disability is, without saying something inappropriate, although I have heard that disabilities are called different abilities, is that right? So, I know very little about disability*

(Faculty 5).

However, despite their lack of specific training or knowledge related to disability, they knew that there was a very varied typology and that the concept of disability was very broad. These faculty members admitted that people with disabilities were those who required additional support for learning. Likewise, they indicated that, in the university, they mostly found people with physical disabilities, and that this type of disability was easier to detect. On the other hand, they also pointed out the existence of other types of disabilities that were not as visible as physical disabilities, e.g., mental disabilities.

*The first thing that comes to my mind is a person who has a physical disability, but that is not the only type of disability, since disability is a broader concept. There are many types of disabilities that can hinder the learning process*

(Faculty 6).

Mainly, the majority of participants conceived disability from two clear theoretical perspectives. Some faculty members presented a conception according to the medical model, whereas others referred to the social model of disability. From the medical perspective, they supposed that people with disabilities had a series of limitations that imposed difficulties and barriers on their daily life.

*A disability is any characteristic of a person that impedes him/her to do certain things that most people can do, with the same conditions, age and training*

(Faculty 5).

The perspective of disability from a social model implies assuming that it is society which generates the needs of people. From this approach, talking about disability involves stigmatising and labelling people. Thus, for these faculty members, any person, under certain circumstances, could have a disability and require adjustments at some point in his/her life.

*Talking about disability, from my point of view, is totally outdated and stigmatising. We must take into account that any person can have specific needs at some point*

(Faculty 2).

Moreover, most of the participants had very little knowledge of pedagogy or educational inclusion. They admitted having no specific faculty training in this regard. In fact, only a few of them had obtained the Pedagogic Aptitude Certificate (CAP, in Spanish) or passed through the new-faculty training of their universities. Others stated that the little knowledge they had originated from their own experience or the exchange of information with other colleagues. However, despite this lack of training, some of them defined disability from an integrative perspective, in which the students with disabilities had to be present, be part of the group-class and develop their sense of belonging.

*I know that the aim is to integrate everyone, work with groups of people in which they all feel integrated and ensure that they are part of the class*

(Faculty 4).

Furthermore, some participants associated the meaning of inclusive education with the acceptance and recognition of differences. In their opinion, inclusive education was not the labelling of a certain group or specific people, but rather a philosophy that had to be present at all stages of education and which implied empathising with the other person.

*Inclusive education is about educating in the acceptance of diversity, promoting empathy. It is fundamental in all these matters to always put oneself in the other person's shoes and help*

(Faculty 21).

One of the key aspects of this study is that the majority of faculty members had very little training and knowledge about disability and inclusion. Nevertheless, they were selected by their students as inclusive lecturers. This assertion leads us to question if disability training is part of the foundations of inclusive practices. What did motivate these faculty members to be inclusive? What reasons or attitudes underlie these inclusive practices?

Data indicate that these inclusive attitudes are related either to previous experiences of most of the participants, both at the workplace and in their personal lives, or their ethical values. They considered that it was an ethical and professional obligation to reach all students and facilitate their learning beyond their disabilities or personal characteristics.

*I do it because it is my duty. My job is to try and reach as many students as possible. There is no greater satisfaction than seeing how you manage to improve the life of a person*

(Faculty 2).

Another motivation was associated with the sensitivity, empathy and personal involvement of each faculty member. Most of the participants asserted that their motivation for providing

an inclusive response to students with disabilities was to reduce the difficulties they could experience in the learning process. At the same time, this was the same motivation that drove them to attend to any other student adequately.

*I like caring about people, regardless of whether or not they have a disability. I try to help them as much as I can*

(Faculty 6).

Finally, some faculty members highlighted that their motivation for carrying out inclusive practices in the classroom was a matter of fundamental rights. These professionals pointed out that universities, and thus faculty members, are at the service of people and should provide the conditions to guarantee equal opportunities as social justice for all students, without exceptions.

*Everyone has the right to the same opportunities. You may have some type of special need, but you do not have to spend your life making twice as much effort as other people to achieve the same goal*

(Faculty 4).

## The actions taken by faculty members as an evidence of their inclusive work

More than half of participants stated that they carried out most of their inclusive practices almost unconsciously, based on goodwill actions and their predisposition to help. These actions were related to applying common sense responding to the specific needs of the students.

The first action consisted in learning more about the circumstances of the students with disabilities. Most of these faculty members delved, on their own, into the requirements of each type of disability and the needs of each student. In general, they did so by directly asking the students what they needed and how they could help.

*First, I get in contact with him/her directly, in order to know more or less how I can help him/ her. Not all disabilities or functional diversities are the same*

(Faculty 3).

Moreover, some of them also searched for further information on how to act in the class-room to attend to certain types of disability. To this end, they consulted the disability services of the university, specialised services outside of the university or colleagues with similar experiences.

*I asked other colleagues what they had done, what they had learned, what assignments they gave to the students so I could link them to my selection of assignments.*

(Faculty 23).

The second action was related to the reasonable adjustments that they carried out to adapt to the needs of the students. The majority of faculty members, despite their lack of training in how to do this, had a predisposition and positive attitude toward making reasonable

adjustments in their subject. They pointed out that it was very important not to 'lower' the content levels of the subject, but to adapt elements such as method (how the contents are taught), materials (type of documents and font size) and even evaluation (exam format and time to complete it).

> *In class, I use presentations, videos and a set of instruments to reinforce certain contents. Then, if there is someone with a special need, obviously, I give him/her the material in advance so that he/she can see the slides better from his/her computer*

(Faculty 1).

The third action was related to follow-up and attention to the students through the tutorial sessions. More than half of faculty members were eager to extend their tutoring schedule if necessary and even to invite some students who seemed to need it. They stated that these tutorials helped them to track the learning process more individually and to delve into possible difficulties that the students could be encountering in the subject.

> *We met once every week for one hour. The students came to see me and told me how he/she was doing with the subject, whether he/she had any doubts or whether he/she had felt frustrated at any point in class*

(Faculty 8).

Finally, a few participants mentioned that they carried out certain practices in the classroom that complemented the previous practices. One of these actions consisted of giving the contents to the students in advance. Another action was the promotion of support among peers. The faculty members highlighted that interaction between students was key for students with disabilities to feel comfortable and to learn more.

> *Being aware of their environment, of the classmates with whom they have more contact, even without the student knowing. There are more advanced students, whom I know are closer to them; so, I make sure they help them, give them lecture notes, etc.*

(Faculty 2)

## Proposals for improving the current situation of students with disabilities

The first suggestion made by all of the participants for improving the situation of students with disabilities in the university referred to making reasonable adjustments in the classroom. Specifically, most of the participants explained that the disability services provided some general guidelines to follow with those students who communicated their needs. Such information, according to the faculty members, was useful in practice for their subject, although it was insufficient for the development of inclusive practices. From their point of view, it was necessary to have better coordination and support from the disability services.

> *They give you some general steps to follow, but this depends a lot on your personal attitude. They only told me that this person did not see well and that the letters had to be big. And what knowledge do I have about people with visual impairment? Nobody has trained me in that matter. So, I have to improvise on the fly.*

(Faculty 11)

The second aspect that most of the participants considered to be in need of improvement was faculty training and information. They expressed, in general, their dissatisfaction with the training received, especially regarding attention to diversity. They stated that nobody had helped them to face the challenge of having a student with a disability in the classroom and providing an adequate response. Nonetheless, they explained that it was essential to have a proactive attitude toward helping each student, ensuring one's training and finding alternatives for oneself.

*Nobody helped me, but I took a course in order to improve both me, in terms of satisfaction, and the students, in terms of results. So, I think every faculty member must be involved and search for those courses.*

(Faculty 1)

Additionally, the majority of participants made a series of suggestions about some aspects that could be improved for a more inclusive university.

One of these suggestions was related to disability services. Some faculty members proposed that there should be more active and committed specialists working jointly with each faculty member to determine which reasonable adjustments should be made.

*In many cases, it is not possible to help students as much as one would like to, simply because one is not aware of these circumstance in the classroom. So, I think that we should work jointly and follow up with the students, see which classrooms they are in, which subjects they are registered for and which guidelines are most adequate to follow*

(Faculty 6).

Another suggestion was focused on faculty training. Most of the participants emphasised that faculty members must be offered comprehensive pedagogical training about inclusion. They believed that it was essential to learn how to make inclusive adaptations for teaching projects and methodologies to be valid for all students.

*The university should coordinate, train and inform better. It would be ideal if all faculty members received a series of guidelines to adapt our projects, our methodology. . .I do this in a self-taught manner and of my own free will*

(Faculty 1).

Another proposal made by some participants was that the university should be designed for every student from the beginning. This proposal was framed within the basic pillars of UDL. They pointed out that it was fundamental to make a mentality change about the meaning of providing an inclusive response, as well as to improve the regulations, followed by a study of the architectural, didactic, political and cultural barriers present in each department. This would promote awareness of how difficulties encountered by students were actually imposed by the institution and not an intrinsic characteristic of people with disabilities.

*The university must take into account students with disabilities from the base of its design. Everything we propose, in any matter (classrooms, workshops, scholarships, resources. . .), must be designed in a way that students with disabilities can have the same opportunities as any other student*

(Faculty 4).

Finally, a few participants considered it necessary to improve the coordination among faculty members and create student support groups in order to work together for the inclusion of all students. From their perspective, the reality of university was individualistic and did not include favourable communication between departments and professionals. Student support groups would improve the academic success of students with disabilities, and teams of faculty members would allow exchanging concerns, doubts and useful experiences to reflect on the practice and improve their teaching.

> *Is there a forum for faculty members to help when there is a student with a disability in class?*
> *I do not know whether my colleague in the office next to mine has a student with a disability.*
> *Nobody knows what I have experienced with my students*

(Faculty 11).

## Discussion

Obtaining the opinions of Social and Legal Sciences faculty members allowed us to understand that, in order to be inclusive, training is relevant, but as other previous studies have already pointed out [8, 31], attitude is even more important, i.e., the good will and eagerness to do one's job properly. The characteristics that define these faculty members, the way they perceive disability and inclusive education and their teaching practices can be the leverage to initiate change and promote more equitable and humane practices.

First, the characteristics that define the profile of the faculty members identified in this study are in line with those reported in previous research [21, 23], which indicates that inclusive faculty members are generally empathetic, communicative, motivating, facilitating and sensitive people who care about establishing affective, close and trust-based relationships with their students. Nevertheless, our study shows additional characteristics of inclusive faculty members of an unexplored field of knowledge. These previously mentioned characteristics are corroborated, although, at the same time, others are included, which have not been mentioned to date, such as responsibility and vocation. According to Florian [28], these participants enjoy their job and feel that this requires them to give the best of themselves to every student every day. Through informative and reflective sessions about the teaching practice, faculty members should become aware of the relevance of their role and the characteristics that define them in order to contribute to the inclusion of all students.

Second, it is significant that many of the participants, despite their lack of pedagogical training, had an open-minded concept of disability and inclusive education. Specifically, most of them did not focus on the hypothetical deficits of the student, but on their own responsibility as educators and on the support that all persons deserve and need, as well as on the barriers that are imposed by the environment and by society [34, 37]. Therefore, the narratives of the participants in this research can contribute to transforming the practices of other faculty members to make them more inclusive. This way, studies such as ours can help to move towards the social model rather than the medical model of disability that is still very present among faculty members and institutions [10, 11]. Although a minority of the participants did not want to explain these concepts, due to their lack of training [31], a large number of them were convinced that disabilities are not within people, but in the means made available to them [38]. In agreement with Bunbury [12], if all universities taught compulsory courses on awareness in disability, confidence of faculty members in their inclusive response would increase and prejudice about disability would decrease.

With respect to the actions that the faculty members took to respond to students with disabilities, our research is in line with previous studies [16, 22] on how to meet the needs of all students, as all faculty members were based on having a positive attitude toward such response, without favourable treatments. In fact, as was described, these faculty members developed inclusive practices and made reasonable adjustments, sometimes unconsciously: "*applying humane treatment and common sense*". As other studies on accessibility in in HE indicated [9, 12, 19], these strategies based on UDL benefited not only the students with disabilities, but actually all students.

In addition to reasonable adjustments in the subject, the faculty members took other actions, such as tutorials, the guidelines of disability services and advice from other colleagues, in order to know the situation of students with disabilities and how to help them [21]. This is not in agreement with other studies [32, 33] which report the inflexibility and little availability of the faculty to make adjustments. Despite their lack of training, the participants did their part and searched for the necessary solutions to provide an inclusive response to their students. In this sense, it would be helpful for faculty members to be trained in how to work from the UDL approach, with the aim of increasing their confidence within their profession and their inclusion of the students in the classroom.

We recognize that this study is not exempt from limitations. Firstly, it would be interesting to compare the opinions of the participants with those of other faculty members of the university community, such as colleagues and administration staff. Secondly, it would be enriching for this investigation to compare the results with those of other universities, both at national and international level.

## Recommendations and future research

Regarding the aspects that could be improved in the inclusion of students, these faculty members highlighted two basic lacks to be covered among the faculty and in the university as an institution. The first one was related to the support provided by the disability services to help faculty members to make the reasonable adjustments according to the type of disability. [26, 39]. Participants pointed out their concerns regarding the lack of coordination between these services, the faculty and the students. This idea already supported by other researchers [34] is backed regarding that limitations, in some cases, are imposed by culture itself and by how the university understands and welcomes disability. It would be adequate to have awareness campaigns about disability, as well as work groups among faculty members of the same department to reflect and share concerns, experiences with students and didactic resources in a collaborative manner. The second one was related to the lack of a firm response by the university with an integral plan of pedagogical training and information about disability [29, 31]. It is clear that faculty members must be proactive, eager to receive training and have initiative, although the opinions of the participants become guidelines for universities to listen to the needs of the faculty and, consequently, take care of and commit to a proper and quality faculty training.

Future research should extend the analysis to the reasons that influence the academic success of students with disabilities and how they experience their inclusion in the labour market.

## Conclusions

In short, the results presented in this study contain a series of lessons that can help Social and Legal Sciences faculty members to feel and understand that they are a key element to achieve the learning, participation and success of their students [20, 21]. The different opinions that are shown in this article about the characteristics of inclusive faculty members, their way of understanding disability and their action, as well as their proposals for improvement, reveal

clues to prevent the violation of the right of people with disabilities in HE. Moreover, university directors must design, develop and evaluate training programmes focused on the conceptions of disability and inclusive practices.

## Supporting information

**S1 File.**
(ZIP)

## Author Contributions

**Conceptualization:** Rosario Lopez-Gavira, Inmaculada Orozco, Ana Doménech.

**Data curation:** Rosario Lopez-Gavira, Inmaculada Orozco, Ana Doménech.

**Formal analysis:** Rosario Lopez-Gavira, Inmaculada Orozco, Ana Doménech.

**Funding acquisition:** Rosario Lopez-Gavira, Inmaculada Orozco, Ana Doménech.

**Investigation:** Rosario Lopez-Gavira, Inmaculada Orozco, Ana Doménech.

**Methodology:** Rosario Lopez-Gavira, Inmaculada Orozco, Ana Doménech.

**Project administration:** Rosario Lopez-Gavira, Inmaculada Orozco, Ana Doménech.

**Writing – original draft:** Rosario Lopez-Gavira, Inmaculada Orozco, Ana Doménech.

**Writing – review & editing:** Rosario Lopez-Gavira, Inmaculada Orozco, Ana Doménech.

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
