## [Decision Letter · Decision Letter 0]

5 May 2021

PONE-D-21-01964

Is pedagogical training an essential requirement for inclusive education? The case of faculty members in the area of Social and Legal Sciences in Spain

PLOS ONE

Dear Dr. Lopez-Gavira,

Thank you for submitting your manuscript to PLOS ONE. After careful consideration, we feel that it has merit but does not fully meet PLOS ONE’s publication criteria as it currently stands. Therefore, we invite you to submit a revised version of the manuscript that addresses the points raised during the review process.

The Academic editor served as the second reviewer for this manuscript and agree with the minor revision recommendations.

We look forward to receiving your revised manuscript.

Kind regards,

Joseph Telfair, DrPH, MSW, MPH

Academic Editor

PLOS ONE

Journal Requirements:

3. Peer review at PLOS ONE is not double-blinded (https://journals.plos.org/plosone/s/editorial-and-peer-review-process). For this reason, authors should include in the revised manuscript all the information removed for blind review.

Reviewers' comments:

Reviewer's Responses to Questions

**Comments to the Author**

1. Is the manuscript technically sound, and do the data support the conclusions?

Reviewer #1: Partly

2. Has the statistical analysis been performed appropriately and rigorously? 

Reviewer #1: N/A

3. Have the authors made all data underlying the findings in their manuscript fully available?

Reviewer #1: No

4. Is the manuscript presented in an intelligible fashion and written in standard English?

Reviewer #1: No

5. Review Comments to the Author

Reviewer #1: Please see attached notes for minor content and writing style/grammar recommendations.

PONE-D-21-01964 Reviewer Comments

Thank you for allowing me to review this valuable work. I appreciate the research questions you aimed to answer. You provide some excellent insight into the perceptions and practices of university faculty regarding students with disabilities. Your work contributes to the current literature and scholarship in this area and helps the field of inclusive education move forward. Below are my recommendations to prepare this manuscript for publication.

Line 51: extra words (to the)

Line 53: “turning into an inclusive faculty member”- maybe replace ‘turning into’ with ‘becoming’

Line 73: should say ‘students with disabilities”

Line 247-248- I think you should end your section with something other than a quotation. Could you include your quotations within the paragraphs rather than always at the end?

In various sentences in the results section you indicate that ‘the faculty members’ indicated something. It would strengthen your results if you could provide some clearer indication of how a result was selected. This could be discussed in the data analysis, perhaps when you talk about the methods of analyzing this data. Out of 25 participants, how many had to mention the term in your analysis software in order to be included as one of the codes? Using terminology such as “most, over half, few, etc. would be helpful. Which you actually do later on.

In the discussion section, making explicit reference to how this research connects to prior research would better support the claims you make, rather than simple listing the references at the end of a statement about your findings.

448-450- I don’t think you can make this claim that these universities do not align with former research reporting a medical model since you have 25 faculty over 7 Universities, who were nominated for being inclusive educators. It would be important to reframe this point so that it better represents how your research fits within the research you are citing that indicates a medical model is still evident in many Universities. I believe that your work plays an important part in moving away from this but I do not believe the findings to be contradictory given your sample of participants.

You make a similar claim again in 464-466 (same comments as above regarding framing this within the limitations of the current study)

473- The first shortcoming was related to the support provided by the disability services to make the reasonable adjustments an adequate manner- this sentence needs revising, perhaps there are a few words missing.

Although you mention a few recommendations for moving forward throughout your discussion and conclusion, this should be a section on its own to highlight your recommendations based on your findings and where you would see research extending on these findings. Then follow with your conclusion of the overall findings as they relate to the bigger picture within the research.

554- who is Frank et al? are you missing some information?

453-455- If you make this claim, you should support it more with a connection to your study. It is a bold statement that training would fix the challenges faced by faculty members, but there is an entire body of research that supports the influence of ones (educators) attitudes and beliefs about disability and opportunities for positive experience with ppl with disabilities that contribute to their inclusive practice.

6. PLOS authors have the option to publish the peer review history of their article (what does this mean?). If published, this will include your full peer review and any attached files.

Reviewer #1: No

---

## [Author Response · Author response to Decision Letter 0]

18 May 2021

Thank you for the review. We appreciate the opportunity to review this manuscript and improve its quality. Foremost, we would like to thank the editor and reviewers for their constructive feedback. We have made significant changes in our work. We apologize for the time it took to send in the revision of the paper, but we wanted to analyze and review in depth each of the comments.

We have completed a revision of this manuscript based on the comments and suggestions given, which we believe has improved the paper.

---

## [Decision Letter · Decision Letter 1]

8 Jun 2021

PONE-D-21-01964R1

Is pedagogical training an essential requirement for inclusive education? The case of faculty members in the area of Social and Legal Sciences in Spain

PLOS ONE

Dear Dr. Lopez-Gavira,

Thank you for submitting your manuscript to PLOS ONE. After careful consideration, we feel that it has merit but does not fully meet PLOS ONE’s publication criteria as it currently stands. Therefore, we invite you to submit a revised version of the manuscript that addresses the points raised during the review process.

There are some very minor changes suggested by a reviewer. The Academic Editor, serving as the second reviewer agrees and the authors are asked to provide these changes for further consideration for publication. 

We look forward to receiving your revised manuscript.

Kind regards,

Joseph Telfair, DrPH, MSW, MPH

Academic Editor

PLOS ONE

Journal Requirements:

Reviewers' comments:

Reviewer's Responses to Questions

**Comments to the Author**

1. If the authors have adequately addressed your comments raised in a previous round of review and you feel that this manuscript is now acceptable for publication, you may indicate that here to bypass the “Comments to the Author” section, enter your conflict of interest statement in the “Confidential to Editor” section, and submit your "Accept" recommendation.

Reviewer #1: (No Response)

2. Is the manuscript technically sound, and do the data support the conclusions?

Reviewer #1: Yes

3. Has the statistical analysis been performed appropriately and rigorously? 

Reviewer #1: N/A

4. Have the authors made all data underlying the findings in their manuscript fully available?

Reviewer #1: Yes

5. Is the manuscript presented in an intelligible fashion and written in standard English?

Reviewer #1: Yes

6. Review Comments to the Author

Reviewer #1: Just a few minor adjustments needed related to language/word choice: Changes in " "

298 This assertion leads us to question whether training is not part of the foundations of inclusive practices.- this sentence is confusing and should consider rewriting for clarity- “This assertion leads us to question if disability training is part of the foundations of inclusive education”

353 Should read “More than half” not Most than half

361 One of these actions "consisted in" giving the contents to the students in advance.- should be “consisted of”

411 change to: This would "promote" awareness of how difficulties encountered by students were actually imposed by the institution and not an intrinsic characteristic of people with disabilities.

509 replaced "to" with "in" improvement, reveal clues to prevent the violation of the right of people with disabilities "in" HE.

Thank you- great work!

7. PLOS authors have the option to publish the peer review history of their article (what does this mean?). If published, this will include your full peer review and any attached files.

Reviewer #1: No

---

## [Author Response · Author response to Decision Letter 1]

15 Jun 2021

Dear Sirs, Thank you for the second review. We have completed a revision of this manuscript based on your comments and suggestions, which we believe has improved the paper for further consideration for publication.

We have included an additional copy of our revised manuscript that does not contain any tracked changes or highlighting as the main article file and we have amended the file type for the file showing your changes to Revised Manuscript w/tracked changes (File date 14th June)

Thank you very much

---

## [Decision Letter · Decision Letter 2]

24 Jun 2021

Is pedagogical training an essential requirement for inclusive education? The case of faculty members in the area of Social and Legal Sciences in Spain

PONE-D-21-01964R2

Dear Dr. Lopez-Gavira,

We’re pleased to inform you that your manuscript has been judged scientifically suitable for publication and will be formally accepted for publication once it meets all outstanding technical requirements.

Kind regards,

Joseph Telfair, DrPH, MSW, MPH

Academic Editor

PLOS ONE

Additional Editor Comments (optional):

Reviewers' comments:

Reviewer's Responses to Questions

**Comments to the Author**

1. If the authors have adequately addressed your comments raised in a previous round of review and you feel that this manuscript is now acceptable for publication, you may indicate that here to bypass the “Comments to the Author” section, enter your conflict of interest statement in the “Confidential to Editor” section, and submit your "Accept" recommendation.

Reviewer #1: All comments have been addressed

2. Is the manuscript technically sound, and do the data support the conclusions?

Reviewer #1: Yes

3. Has the statistical analysis been performed appropriately and rigorously? 

Reviewer #1: N/A

4. Have the authors made all data underlying the findings in their manuscript fully available?

Reviewer #1: Yes

5. Is the manuscript presented in an intelligible fashion and written in standard English?

Reviewer #1: Yes

6. Review Comments to the Author

Reviewer #1: Thanks for all the reviews made. The paper reads well and presents important information on the topic of attitudes towards inclusion at post secondary institutions.

7. PLOS authors have the option to publish the peer review history of their article (what does this mean?). If published, this will include your full peer review and any attached files.

Reviewer #1: No

---

## [Editor Report · Acceptance letter]

25 Jun 2021

PONE-D-21-01964R2 

Is pedagogical training an essential requirement for inclusive education? The case of faculty members in the area of Social and Legal Sciences in Spain 

Dear Dr. Lopez-Gavira:

I'm pleased to inform you that your manuscript has been deemed suitable for publication in PLOS ONE. Congratulations! Your manuscript is now with our production department. 

Kind regards, 

on behalf of

Dr. Joseph Telfair 

Academic Editor

PLOS ONE